# Impact of the "Looking after my health after cancer" peer-led active patient education program on cancer survivors and their caregivers: A qualitative study

Ainhoa Ulibarri-Ochoa[1,2,3]*, Sheila Sánchez-Gómez[4,5,6], Estíbaliz Gamboa-Moreno[7,8,9], Irene Duo-Trecet[7,10], Lucia Garate-Echenique[11], Begoña Belarra-Tellechea[12,13], Lourdes Ochoa de Retana-García[7,10]

1 Bioaraba Health Research Institute, Clinical Nursing and Community Health Research Group, Vitoria-Gasteiz, Spain, 2 Osakidetza Basque Health Service, Araba Integrated Health Organisation, Vitoria-Gasteiz, Spain, 3 Vitoria-Gasteiz School of Nursing, University of the Basque Country, Vitoria-Gasteiz, Spain, 4 Bioaraba Health Research Institute, Healthcare Research Group, Vitoria-Gasteiz, Spain, 5 Osakidetza Basque Health Service, Araba Integrated Health Organisation, Directorate for Healthcare Integration, Vitoria-Gasteiz, Spain, 6 Osakidetza Basque Health Service, General Directorate, Nursing Subcommittee, Araba Multidisciplinary Teaching Unit for Family and Community Healthcare, Vitoria-Gasteiz, Spain, 7 Biodonostia Health Research Institute, Primary Care Research Unit, San Sebastián-Donostia, Spain, 8 Osakidetza Basque Health Service, Donostialdea Integrated Health Organisation, San Sebastián-Donostia, Spain, 9 Network for Research on Chronicity, Primary Care, and Health Promotion (RICAPPS), 10 Osakidetza Basque Health Service, Active Patient Program (*Paziente Bizia-Paciente Activo*), Subdirectorate for Primary Care Coordination, San Sebastián-Donostia, Spain, 11 Osakidetza Basque Health Service, Coordination of Strategic Plans, Subdirectorate for Nursing, Vitoria-Gasteiz, Spain, 12 Biocruces Bizkaia Health Research Institute, Barakaldo, Spain, 13 Osakidetza Basque Health Service, School of Health (*Osasun Eskola*), Subdirectorate for Primary Care Coordination, Vitoria-Gasteiz, Spain

* ainhoa.ulibarriochoa@osakidetza.eus

**Data Availability Statement:** All relevant data are within the paper and the figures.

## Abstract

### Background

Cancer survival has doubled and is likely to continue increasing in the near future. Cancer survivors experience long-term adverse effects, with associated psychological changes, and often they have needs that are yet to be met. Recognizing the lack of continuity-of-care initiatives for cancer survivors and caregivers, Osakidetza Basque Health Service has started to implement through primary care a peer-led active patient education program called "Looking after my health after cancer". This study explores how cancer survivors and their caregivers rate the experience of participating in the program, to what extent the program helps them understand and address their unmet felt needs, and helps them improve their activation for self-care and self-management.

### Methods

A qualitative exploratory phenomenological study was conducted using five focus groups: four with cancer survivors (n = 29) and caregivers (n = 2), and one with peer leaders (n = 7). Narrative content analysis was performed using the constant comparison method, facilitated by Atlas-ti software. Descriptive analysis of sociodemographic and clinical data was

**Funding:** The study was awarded the III Research Prize for Nursing in Cancer and Palliative Care from the Angel Muriel Foundation. Further, a grant to support nursing research was received from the College of Registered Nurses of Gipuzkoa (COEGI 2019). Likewise, with funding from Biodonostia Health Research Institute, the manuscript has been translated and reviewed by the scientific editors of Ideas Need Communicating Language Services. The funders had no role in study design, data collection and analysis, decision to publish, or preparation of the manuscript.

**Competing interests:** The authors have declared that no competing interests exist.

performed. The study was developed according to the Consolidated criteria for reporting qualitative research (COREQ) checklist.

## Results

Five main themes emerged from the content analysis: 1) satisfaction with the program as a positive learning experience; 2) peer sharing and learning ("if they can, so can I"); 3) fears prior to attending the program; 4) becoming more aware of unmet felt needs and feeling understood in the "new normal"; and 5) a more positive view of their experience, helping them become active in self-care and empowered in the self-management of their condition.

## Conclusions

The peer education program has shown to have a positive impact on cancer survivors and caregivers. It is necessary to design, implement and evaluate interventions of this type to address unmet felt needs during cancer survivorship and improve their quality of life.

## Introduction

According to the latest results published by the *International Agency for Research on Cancer*, cancer remains one of the largest public health problems, with around 19 million new cases and 10 million deaths worldwide [1] and more than 282,000 new cases and 113,000 deaths in Spain [2]. On the other hand, cancer death rates have decreased significantly in recent decades due to improvements in prevention and early diagnosis, and therapeutic advances. Survival has doubled over the last 40 years in Spain and is likely to continue increasing in the near future, leading to a growing number of cancer survivors [3].

The *National Cancer Institute* (NCI) states that survivorship concerns "*the health and well-being of a person with cancer from the time of diagnosis until the end of life. This includes the physical, mental, emotional, social, and financial effects of cancer that begin at diagnosis and continue through treatment and beyond. The survivorship experience also includes issues related to follow-up care (including regular health and wellness checkups), late effects of treatment, cancer recurrence, second cancers, and quality of life. Family members, friends, and caregivers are also considered part of the survivorship experience*" [4]. Survivorship can be described in terms of three phases: (1) acute survival, from diagnosis until the end of the initial treatment; (2) extended survival from the end of the initial treatment continuing through the following months; and (3) permanent survival, identified as cure or prolonged survival after remission [5].

During survivorship, individuals face a "new normal" to which they have to adapt and they ask themselves how their life is going to be from now on. After the end of the active treatment, they undergo various changes, some of which last for several years and others become permanent. Survivors and their next of kin have to face a reality that is sometimes very different from their lives before cancer [6]. They may experience numerous challenges after the treatment, including physical changes such as long-term fatigue; emotional changes such as fear of recurrence; and practical changes in daily life such as starting back at work [7, 8].

The report of the Spanish Group of Patients with Cancer (*GEPAC*) on the needs of cancer survivors notes that they experience long-term adverse effects, with associated psychological changes, and often they have needs that are yet to be met [8]. These unmet needs have also been documented in recent studies [9] and highlighted in declarations from scientific societies

such as the Spanish Oncology Nursing Society (*SEEO*) [10]) and the Spanish Society of Medical Oncology (*SEOM*) [11].

Although progress has been made in identifying the needs of cancer survivors since the report published by the US Institute of Medicine, "From Cancer Patient to Cancer Survivor: Lost in Transition" [12], there is still a marked lack of comprehensive care for this population, there being a need to create a care model for survivors that promotes greater continuity of care and coordination between care levels with individualized follow-up plans [13]. In Spain, the Cancer Strategy of the National Health System also highlights the need to improve coordination between care levels, by developing standardized and coordinated pathways between primary and specialized care to optimize care for cancer survivors, as well as the need to produce individualized follow-up plans, maintain quality of life, maximizing the recovery of functional capacity, and promote healthy lifestyles [14]. Scientific literature in this field highlights the need to implement training programs for cancer survivors to encourage healthy lifestyles and establish follow-up plans that promote full rehabilitation, addressing the physical, psychological, and social factors, and considering the need for support in the management including self-management of adverse effects that may last beyond the end of treatment, all this with the final goal of addressing the needs of this group of patients and improving their quality of life [15, 16].

There is evidence that the participation of cancer survivors in education programs can improve their quality of life. A pre- and post-intervention study in the USA, based on a 6-week psychoeducational program for cancer survivors and their family caregivers, showed improvements in physical, emotional, and functional quality of life in both survivors and caregivers [17]. In line with this, another education program, in this case in Denmark, showed improvements in self-management of care in breast cancer survivors [18]. International scientific literature highlights the value of peer support in education programs for cancer survivors such as the Chronic Disease Self-Management Program [19] and the Expert Patient Program [20], through which, thanks to peer support, patients and their caregivers become better able to understand their illness, take responsibility for their own health and properly manage treatment options.

Recognizing the lack of continuity-of-care initiatives for cancer survivors in Spain in general and the Basque Country in particular, and based on the Chronic Disease Self-Management Program [21], a model developed at the University of Stanford, Osakidetza (the Basque Health Service) has designed and started to implement through primary care a peer-led active patient education program called "Looking after my health after cancer" [22]. This program is part of the 2018–2023 Basque Plan for Cancer that reflects the need to contribute to promoting healthy lifestyles in cancer survivors to achieve better outcomes during their recovery process, as well as provide integrated care that addresses their needs [23].

As with any intervention implemented for the first time, its contribution and impact need to be assessed. Therefore, the objectives of this study were: 1) to explore how cancer survivors and their caregivers rate their experience of participating in the "Looking after my health after cancer" peer-led active patient education program; 2) to assess to what extent participation in the program helps them understand and address their unmet felt needs; and 3) to assess to what extent they consider that the program helps patients enhance their activation for self-care and self-management of their condition.

Taking into account the need to provide continuity of care to a growing number of cancer survivors with complex needs, this program offers an opportunity to promote the acquisition of skills for self-management and self-care, that could be expected to result in improvements in quality of life and greater participation in health-related decision-making.

## Materials and methods

### Study design

It was conducted a qualitative study with a phenomenological design using focus groups. Understanding health promotion and education to be complex activities, it was considered that evaluation of the program through the experience of participants in the educational sessions, in essence, a holistic qualitative approach, would provide a key and enriching view of this reality, that is necessary to understand, transform and improve it. Focus groups were chosen as the primary strategy for the generation of data given that they offer an opportunity for interaction between peers (people who have been through the same experience), which facilitates the gathering of detailed information concerning the collective experience [24]. The study was developed according to the Consolidated criteria for reporting qualitative research (COREQ) checklist [25].

### Description of the peer education program

The "Looking after my health after cancer" active patient program (*Paziente Bizia-Paciente Activo*) is carried out with a focus on peer support and learning, its goal being to address the unmet felt needs of cancer survivors. It seeks to empower these people and help them acquire skills and tools for activation in self-care and self-management of their own health with the goal of improving their quality of life.

The program is for individuals and/or caregivers who have gone through cancer at least 1 year after active treatment, and hence, are in the extended or permanent survival phase. It consists of a 2.5-hour session once a week for 7 weeks, with a mean group size of 12–15 people, led by 2 peer leaders who are themselves cancer survivors or main caregivers of cancer survivors and have previously been specifically trained for the program (Fig 1).

The content of the program was designed taking the structure of Stanford's Chronic Disease Self-Management Education Program as a reference. To contextualize it to the care of cancer survivors, a literature review was conducted to identify potential felt unmet needs in this population. A key study was the Spanish Group for Cancer Patients (GEPAC)'s *Report on the needs of cancer survivors* based on a study conducted in Spain a few years earlier. The content of the materials for the various sessions of the program was developed by a group composed of two oncologists, a psychiatrist, a psychologist, two oncology nurses, a nurse responsible for the Patient Care Service, a management nurse, and two primary care doctors and three primary

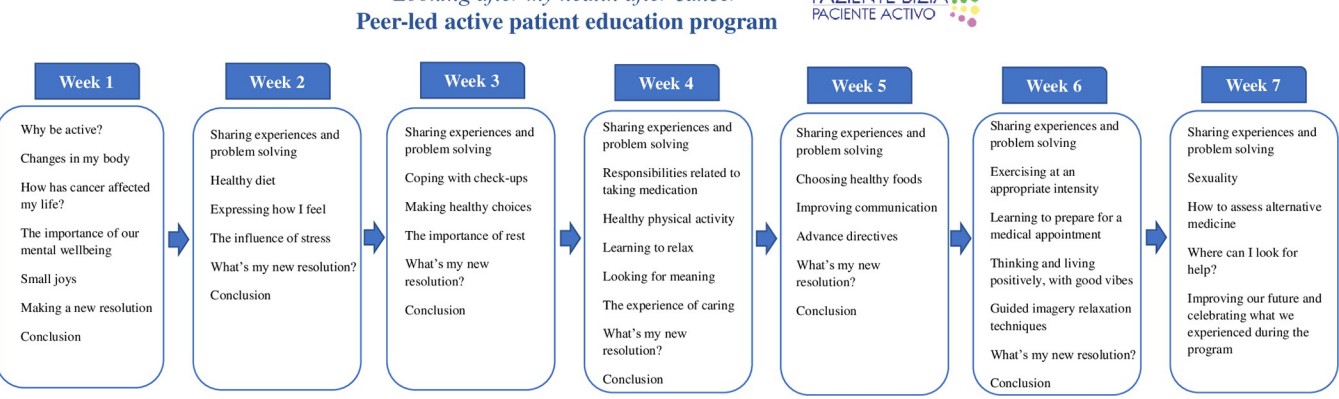

**Fig 1. Structure and content of the "Looking after my health after cancer" peer-led active patient education program.**

care nurses trained in the program's methodology. The materials developed by this group were reviewed by three patients (who were to be program peer leaders) and two patient associations (Spanish Association Against Cancer and Association of Women Affected by Breast and Gynecological Cancer of Gipuzkoa). With their contributions, various materials were reworked. Finally, the program was piloted in a group of seven cancer survivors and feedback was requested after each session. Having completed this first pilot of the program, the content manual was finalized, considering all the comments and changes suggested, and the program was validated.

## Recruitment of participants

Cancer survivors and their caregivers who were ≥ 18 years of age, for whom at least 1 year had passed since the end of the active treatment (that is, they were in the extended or permanent survival phase) and who had attended at least five out of the seven sessions of the education program were included. Peer leaders were also included to triangulate and strengthen the findings.

Participants were selected by theoretical intentional sampling [26]. The researchers sought to obtain a heterogeneous sample by including men and women of different age ranges and with experience related to different types of cancer. The recruitment was performed by the coordinators of the active patient program.

## Data collection

Sociodemographic data concerning age, sex, level of education, occupational status, and social and family circumstances were collected, as well as clinical data such as the type of cancer the survivor had had and the time from diagnosis to joining the program (to estimate the survival phase).

During 2019, the researchers held four focus groups with cancer survivors and caregivers, with between 7 and 9 people in each group. In 2020, a complementary focus group was organized with peer leaders, this being held over Zoom due to the coronavirus disease 2019 pandemic. Focus groups were moderated by various members of the research team who were aware of how the active patient program works and had good communication skills (AU-O, SS-G, LG-E, BB-T, and LO-G). Focus groups were conducted using a semi-structured script composed of open-ended questions (Table 1), drafted to elicit discussion relevant to the study objectives. The mean duration of the focus groups was 60 minutes.

**Table 1. Semi-structured guide consisting of open-ended questions.**

| |
|---|
| • How do you rate your participation in the "Looking after my health after cancer" peer-led active patient education program? Was it what you expected? To what extent has it met your expectations at the outset? |
| • To what extent do you feel it has helped you address your current care needs? |
| • To be more specific, to what extent do you feel it has helped you (or the care recipient) adopt healthy lifestyle habits, manage symptoms, address the impact of having had cancer, cope with emotions, face issues concerning sexuality, develop skills for managing your (their) health, develop skills for communicating with those around you, etc.? |
| • Which features [of the program] and sessions do you consider the most useful? Was there anything you felt was missing? What would you change? |
| • To what extent do you feel that participation in this program has helped you cope with your current condition? To what extent do you feel that participation in this program has helped you be more proactive in your self-care? Did you experience any difficulties participating in the program? |
| • Would you recommend this program to other people? |
| • Are there any issues related to your experience with this program that you think we should know about that we have not yet touched on, that you consider important and that you would like to add? |

## Data analysis

Analysis of themes or narrative content was performed using the constant comparison method proposed by Glaser and Strauss [27]. This type of analysis makes it possible to identify, organize and analyze data in detail and uncover patterns or themes based on careful reading and re-reading of the information collected, and thereby, make inferences that help the researchers to properly understand the phenomenon under study [27]. The cancer survivor and caregiver focus groups were audio-recorded while the peer leader focus group was recorded using Zoom. All the focus groups were transcribed verbatim. Atlas-ti software was used to store data and facilitate the data analysis.

These data were analyzed using phenomenological methods following a simplified version of Hycner's process [28] which has five steps/stages: 1) Bracketing and phenomenological reduction: with an attitude of openness to any meaning that emerged, two researchers repeatedly read the information provided by the key informants in focus groups to become familiar with the testimonies, identify units of meaning and develop a holistic sense of what was expressed by the survivors, caregivers, and peer leaders. Each focus group was identified by a numerical code (from 1 to 5), and in turn, each testimony in each focus group was also coded; 2) Delineating units of meaning: units of relevant meaning were extracted and reviewed, any redundant units being eliminated; 3) Clustering of units of meaning to form themes: the units of meaning were grouped to form themes reflecting the participants' experiences; 4) Summarizing of each focus group, validating the themes and where necessary modifying them: a summary was produced containing all the themes generated from the units of meaning. A validity check was then performed, going back to the informants' testimonies to assess whether the essence of the testimonies in the focus groups had been captured correctly and modifying the summary as appropriate; and 5) Extracting general and unique themes from all the focus groups and making a composite summary: themes common to most of the focus group testimonies were identified, as well as individual variations, and the analysis was concluded by writing a composite summary that reflected the transformation of the everyday expressions of the study participants into expressions appropriate to the scientific discourse supporting the objectives of this research.

Regarding the concept of reflective practice, it is important to point out that one of the two researchers who carried out the initial thematic analysis had first-hand experience of the reality of cancer survivorship as a professional and as a caregiver. This experience helped to enrich the reflective and analytical process and the interpretation of the data that inform the research question. It is also important to note that the two researchers who performed the analysis were trained in qualitative methodology. Further, the other researchers who corroborated the analysis were nurses who were familiar with the education program and the reality of cancer survivors and caregivers.

## Rigor criteria

It was carried out rigorous triangulation of the entire data analysis process using several different researchers. Firstly, two researchers (AU-O and SS-G) trained in qualitative methodology carried out the content analysis. This analysis was then shared with the rest of the researchers, some of whom had moderated the focus groups (LO-G, LG-E, BB-T), while others were familiar with the program (EG-M and ID-T). In this way, the analysis was triangulated among all the researchers.

Further, the focus group with peer leaders enabled the researchers to triangulate the findings from the focus groups with program participants. Given that the peer leaders had also previously participated as cancer survivors or caregivers in the program and had subsequently

been rigorously trained to run the sessions of this program, the researchers believed it important to complement the perspective of participants with that of the peer leaders. In particular, they were the people who oversaw the program and could potentially provide relevant information about what happened in the sessions and whether what was expressed by the participants agreed with their impressions as peer leaders, and thus be able to compare, corroborate and enrich the findings.

### Ethical considerations

The study was approved by the Drug Research Ethics Committee of the Basque Country (CEIm-E, reference number: PI2019144). All participants were informed about the study and provided written informed consent in accordance with the Regulation (EU) 2016/679 on data protection and Spanish Organic Law 3/2018, of 5 December, on the protection of personal data and guarantee of digital rights. Recordings were destroyed on completion of the transcription, in which participants were anonymized.

## Results

### Participants

Five focus groups were held with a total of 38 key informants (4 focus groups with 29 cancer survivors and 2 caregivers; and a fifth focus group with 7 peer leaders). These participants had a mean age of 59.13 years; the majority were women (68.4%) and almost half had experienced breast cancer (47.2%). A high percentage (60.5%) lived with a partner of similar age. Table 2 summarizes the sociodemographic and clinical characteristics of the study participants.

Before participating in the focus groups, participants had attended the education program in 2017 (2 peer leaders, 5.26%), 2018 (17 people, 44.7%, of which 5 were peer leaders), or 2019 (19 people, 50%). The researchers considered that they had reached data saturation with these five groups.

### Impact of the education program on cancer survivors and their caregivers

In relation to the study objectives, five main themes emerged from the content analysis of the five focus group sessions. Specifically, in relation to the first objective, three main themes emerged: 1) satisfaction with the program as a positive learning experience; 2) peer sharing and learning: "*If they can, so can I*"; and 3) fears prior to attending the program. Concerning the second objective, a fourth theme emerged: 4) becoming more aware of and addressing unmet felt needs and feeling understood in the "new normal". And finally, regarding the third objective, a fifth main theme emerged: 5) a more positive view of their experience, helping them become active in self-care and empowered in the self-management of their condition (Table 3).

Fig 2 is a category tree illustrating the main themes and subthemes that emerged from the analysis.

### Satisfaction with the program as a positive learning experience

Cancer survivors who participated in the "Looking after my health after cancer" program reported being satisfied with their participation in the education program with comments such as "*the workshop helps*" (female survivor, 1.95); "*I was really satisfied with the course*" (male survivor, 2.114); "*I left very satisfied*" (female survivor, 2.118); "*we all came in through the door smiling*" (female survivor, 3.50); and "*I really liked it*" (female survivor, 4.24). They also said that the learning experience had been very positive: "*I have really learned a lot*"

**Table 2. Sociodemographic and clinical characteristics of participants.**

| | | n = 38 |
|---|---|---|
| **Sociodemographic characteristics** | | M (SD) |
| **Age, years** | | 59.13 (11.16) |
| | | n (%) |
| **Sex** | | |
| | Female | 26 (68.4) |
| | Male | 12 (31.6) |
| **Type of participant** | | |
| | Cancer survivor | 29 (76.3) |
| | Main caregiver | 2 (5.3) |
| | Peer leader | 7 (18.4) |
| **Level of education** | | |
| | Primary/lower secondary | 13 (34.2) |
| | Technical and vocational | 11 (28.9) |
| | Higher secondary | 4 (10.5) |
| | University | 10 (26.3) |
| **Occupational status** | | |
| | In paid work | 13 (34.2) |
| | Absolute/total incapacity for work | 6 (15.8) |
| | Retired | 17 (44.8) |
| | Unemployed | 1 (2.6) |
| | In unpaid work | 1 (2.6) |
| **Social/family circumstances** | | |
| | Lived with partner of similar age | 23 (60.5) |
| | Lived with family and/or partner and has some degree of dependence | 1 (2.6) |
| | Lived with family, with no physical/psychological dependence | 10 (26.3) |
| | Lived alone with no children or children living far away | 1 (2.6) |
| | Lived alone with children living nearby | 2 (5.20) |
| | Unknown | 1 (2.6) |
| **Clinical characteristics** | | |
| **Type of cancer** | | |
| | Cervix | 1 (2.8) |
| | Colon and rectum | 5 (13.9) |
| | Esophagus | 1 (2.8) |
| | Stomach | 1 (2.8) |
| | Blood | 5 (13.9) |
| | Breast | 17 (47.2) |
| | Prostate | 1 (2.8) |
| | Lung | 2 (5.5) |
| | Kidney | 2 (5.5) |
| | Unknown | 1 (2.8) |
| **Time from diagnosis to program participation** | | |
| | Between 1 and 5 years ("extended survival") | 22 (57.9) |
| | More than 5 years ("permanent survival") | 15 (39.5) |
| | Unknown | 1 (2.6) |

(female survivor, 1.133); "*things come up, of course, they do, everything comes up. One's own stuff, other people's, everything comes out, and in the end, well, it did me good*" (male survivor,

**Table 3. Relationship between the study objectives and themes that emerged in the focus groups.**

| Objectives | Five main themes identified in the content analysis |
|---|---|
| **Objective 1:** To explore how cancer survivors and their caregivers rate their experience of participating in the "Looking after my health after cancer" peer-led active patient education program. | **Main theme 1:** Satisfaction with the program as a positive learning experience. |
| | **Main theme 2:** Peer sharing and learning: "If they can, so can I". |
| | **Main theme 3:** Fears prior to attending the program. |
| **Objective 2:** To assess to what extent participation in the program helps them understand and address their unmet felt needs. | **Main theme 4:** Becoming more aware of and addressing unmet felt needs and feeling understood in the "new normal". |
| **Objective 3:** To assess to what extent they consider that the program helps patients enhance their activation for self-care and self-management of their condition. | **Main theme 5:** A more positive view of their experience, helping them become active in self-care and empowered in the self-management of their condition. |

3.7); and "*it was a very good experience*" (female survivor, 4.8). For these reasons, they added that it was important to publicize and recommend this program to reach and help more people: "*the topic is very important because through many [patients'] associations, word can be spread that the active patient courses are very good and help people*" (female survivor, 4.72).

Peer leaders confirmed the need for and importance of this education program as part of continuity of care for cancer survivors: "*this is another need, as important as good chemotherapy and radiotherapy, the period after is as important as that*" (female leader, 5.137). They also agreed that participation in this program was a positive learning experience for participants: "*I have the feeling that people are very happy; after the course, they are very satisfied*" (male leader, 5.13); "*it's a way of learning to live, and I mean really live*" (male leader, 5.138); and "*you*

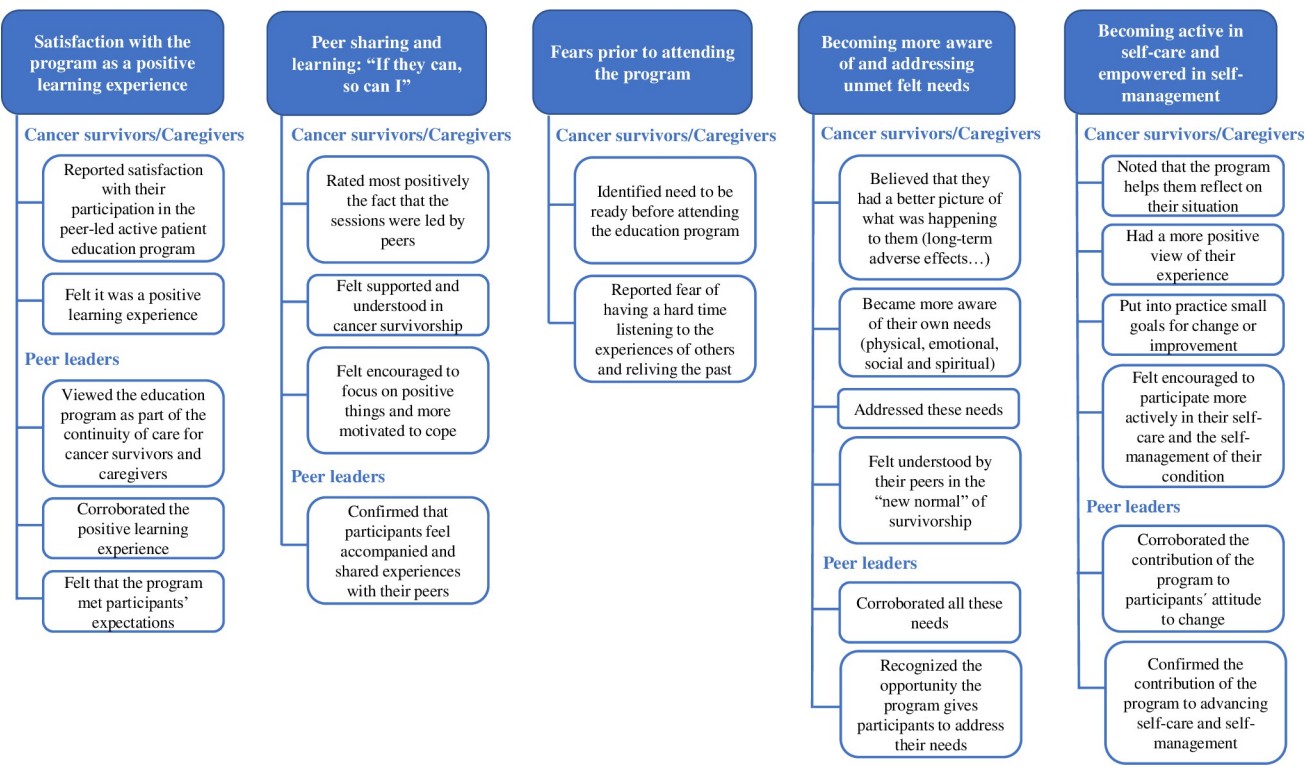

**Fig 2. Category tree.**

*become much more aware of things; from then on, you start learning, and you learn from others, and I believe that there is a good atmosphere in the groups, you always pick that up; you learn this positive attitude*" (female leader, 5.76). Additionally, they highlighted that the education program meets the expectations of participants, noting that "*when on the last day, you go back over the list of the first day, most people have more than met their expectations*" (female leader, 5.26).

### Peer sharing and learning: "If they can, so can I"

One of the elements most positively rated by cancer survivors who participated in the program was that the sessions were led by peers; that is, by people who had been through a similar experience and who had first-hand knowledge of their situation, highlighting that "*they knew more from practical experience, because they had either had cancer themselves or been caregivers of relatives who had and therefore they were people who understood us really well; because what's written in a book, what you get from studying or certain professional training is not the same, it's all theory. They have been through it*" (female survivor, 3.51); and "*it's different explaining a process if you have experienced it rather than just learned about it*" (female survivor, 4.140).

Similarly, they stated that being able to listen to others in the same situation and sharing experiences in a group, in sessions led by peers, made them feel supported and understood in survivorship: "*one of the most positive things about these groups is that you talk and the people listening to you understand you perfectly, because they have been through exactly the same as you*" (male survivor, 2.5); "*somebody who has been through something similar to you understands you more easily and it is like easier to achieve*" (male survivor, 2.72); and "*it´s not easy to talk about this with people who have not been through it; you need people who have been through it, who have suffered like you, for them to really understand you, because others who have been beside you have been with you, helped you, accompanied you, but have not experienced your feelings, your emotions, what goes through your body, through your head, the effects of chemotherapy, the effects of the radiotherapy, they don´t have a clue, only you and people who have lived it know*" (female survivor, 3.11).

In turn, feeling listened to and understood encouraged them to focus on the positive things and feel more motivated to cope with the situation: "*the group itself pulls you up; that is, to see that other people achieve their own goals, this also somehow motivates you*" (male survivor, 3.73); and "*we each have our own reality . . . you cannot see it at that time because you are in another sphere and simply the fact that they give you advice that helps you or support you, now you see it in another way*" (female survivor, 3.76).

Peer leaders confirm that participants seek to obtain information, feel accompanied and share experiences with other people who have been through a similar situation: "*many people seek information, company, to talk about our things*" (female leader, 5.16); and "*a lot is verbalized, a lot is shared, you strongly identify with people you think you are very different from, but who nonetheless have similar feelings or similar behaviors; so the truth is that I believe that it is very enriching to have a group in which you can express yourself freely, truthfully, which is not always the case, because, in one's close circle, one may make people suffer, or worry, so to have something like this seems very useful to me, very enriching and sharing between peers is a key element, the basis of the workshop*" (female leader, 5.53).

### Fears prior to attending the program

Despite rating the program positively, some participants indicate the importance of being ready before attending it, as they reported fearing they would have a bad time listening to the experiences of others and reliving the past: "*if you go earlier, there might be things that you*

*have not considered at all and you suddenly you are faced with them, I think you have to be ready. This fear I had about what I might hear; you have to be ready to cope with it*" (male survivor, 2.101); and "*one [person], another, and all that might stuff drags you down and I also thought I didn't fancy getting dragged down*" (male survivor, 3.17).

### Becoming more aware of and addressing unmet felt needs and feeling understood in the "new normal"

Cancer survivors who participated in this program reported that dealing with the content of the sessions together with people who had gone through a similar process helped them to get a better picture of what was happening to them, identify the long-term adverse effects, become more aware of their own needs, address to these needs and feel understood by their peers in the "new normal" of survivorship.

Below, the researchers summarize the felt needs of participants at physical, emotional, social, and spiritual levels, needs they were able to externalize in the program, becoming more aware of them, and finding answers by addressing them together with their peers.

At the physical level, participants described the enduring nature of late physical adverse effects and how these had an impact on their daily lives. They generally reported feeling tired and weak: "*now I am chronically tired*" (male survivor, 2.87); "*physically you get much weaker*" (male survivor, 2.89); "*there are physical changes. Above all, you feel much less strong*" (2.93), and additionally, there other changes that also affect them in their day to day: "*the symptoms after the treatment, tiredness, etc. . . . it´s everything; it has constrained us, muscle damage, tiredness, painful joints, changes, hair loss*" (male survivor, 3.46); "*you have some memory problems, sometimes you forget things*" (female survivor, 3.82), and make it difficult for them to have a normal life: "*patients like us that have gone through this, we cannot live a completely normal life*" (female survivor, 3.81). All these changes and physical challenges make them feel more fragile: "*I´m feeling things I didn´t before, I´m going downhill*" (male survivor, 2.95).

Further, they reported a need to have more information about their condition: "*so we need more information, we need health professionals to give us that information. Not all the time, but, well, once or twice a year. . .*" (female survivor, 4.128); and a closer follow-up by clinicians to address their needs: "*I think we should have a sort of contact person, who might not be the family doctor, they could be a nurse, a supervisor, it could be someone who does our check-ups . . . we need someone who can address this for us*" (female survivor, 4.92).

At the emotional level, participants reported feeling worried and unsettled in the survivorship stage: "*I look well. . . but . . .*" (male survivor, 1.39); and "*and now what is going to happen to me?*" (female survivor, 1.49). They also expressed difficulties coping with the "new normal", feeling anxious and finding it difficult to cope with the new situation: "*I suppose I need to settle down; I have to take it easier. I have realized I have to go little by little, but it's one thing is to realize it and quite another to accept it. I think my head does not accept it. I want to but I don´t*" (male survivor, 2.96). One of the greatest concerns they verbalized was a fear of recurrence: "*it´s worrying; they have cured my cancer, but the fear. . .*" (female survivor, 3.107). And therefore, they described that one of the most difficult times was when they had to go for check-ups: "*in the first years, that's what it is, you feel stoked. But then, the years go by and the side effects emerge and you are still as frightened of the check-ups as in the first year; a week before the check-up, it pains me. . ., I´m carrying a burden, you can't imagine*" (female survivor, 3.60); and "*it is true that a week before you say 'oh no, I have a check-up', are the test results going to be okay, or perhaps. . .?*" (female survivor, 3.106). Another aspect they mentioned is that they often feel "*misunderstood, which also makes one feel unimaginably powerless*" (female survivor, 3.45). In line with this, participants reported a need to let off steam and that attending this

program helped them share their feelings and concerns and manage their emotions: "*there is a point when you have a need, to say you want to know something, 'who can help me? who can I talk to?' and I mean talk. There comes a point when you don´t know what [to do], you don't say anything as you don't have anyone to talk to. And that's the time for something like this . . . for discussing with other people problems they've had*" (male survivor, 2.102).

In this context, caregivers indicated the need to share their feelings and emotions with other people who may have been in a similar situation, highlighting that it helped them to understand what they had experienced better and connect more closely with their loved ones: "*and when you have been so close to death and what is more that of someone like a child that you love so much, that's the best thing that's happened to you, I needed to rebuild myself and be with other people who had gone through this, talk to other people; then you understand things that might have gone through your head and theirs as well*" (female caregiver, 3.21). The idea of looking after the caregiver is another feature of this program that was considered positive: "*everybody asks the patient, 'How are you doing? How are you?' but nobody cares about the caregiver. Then, one of the sessions was about that, looking after the caregiver, which was very good*" (female caregiver, 2.68).

At the social level, one of the most common feelings reported by participants was loneliness, "*only literally, nobody understood me, because nobody can put themselves in my shoes*" (female survivor, 3.16), since on many occasions they found it difficult to talk to their close circle: "*my husband and my daughter, of course, as you see that they are affected by it almost more than you are, you put on a braver face. Me, no way, I've never cried, to protect them*" (female survivor, 1.57); and this made it difficult to get their support and understanding. For these reasons, they felt that participating in the program was very useful for sharing their emotions, obtaining support and being able to express themselves more freely: "*you need people who are able to understand you*" (female survivor, 3.14); and "*you can talk in a relaxed way; you let off steam, you learn a lot, and talking, you get a lot out*" (female survivor, 4.26).

Participants also mentioned the spiritual side, highlighting the need to work on redirecting and finding purpose in their lives: "*the thing is that now we find ourselves alive in a sort of space that is unfamiliar territory, because it is not one's own mental, physical, or emotional space, nor one's life purpose, because there is a spiritual part that concerns the purpose of life, and 'Why am I stalled here?' Everything is upturned and shaken up*" (female survivor, 3.19). Together with this, some participants highlighted that being through this experience had led to personal growth and an improvement in their self-esteem: "*I see that all of what I have been through has done me some good. It has done me good in that I have improved my self-esteem*" (male survivor, 2.65); and "*this experience has been positive . . . I love myself more. Before I neglected myself; I now have better self-esteem and look after myself more*" (male survivor, 2.66). And this had helped them to value people around them more: "*sometimes you don´t value the person you have in front of you . . . what has happened has done me good because it's made me think and made me more aware of everything*" (male survivor, 2.69); as well as better accept the disease and its consequences: "*you become aware of your disease*" (male survivor, 3.85); and "*better understand the disease*" (female survivor, 4.129).

Peer leaders corroborated all these needs expressed by participants, placing particular emphasis on the fear of recurrence and the program's approach to this issue: "*I have had recurrence; and so I explain this to the participants in the workshops so that they can face it with a certain degree of optimism*" (male leader, 5.66); and "*they see that in the group there may be people who have experienced recurrence; they see that people carry on with their lives, that they do exercise, that they make plans, that they travel, they live . . . in a way, it prepares you*" (female leader, 5.67). They also affirmed the major contribution of the program to helping participants manage their emotions: "*I believe that the workshop after cancer helps you most emotionally; the*

*emotions are what it most draws out, and I think this is the most beautiful thing*" (female leader, 5.56); and "*being able to share your emotions, in my opinion, this is the greatest need; it is the cornerstone, I would consider it the cornerstone*" (female leader, 5.60). In relation to this, peer leaders confirmed that participants found it difficult to share their feelings and needs with people around them: "*people feel really lonely, if you are lame, you go to a bar and you tell your friends, but if you have cancer, you don´t tell anyone, you keep it to yourself*" (male leader, 5.31); and they also highlighted the opportunity the program gives participants to express themselves, share their emotions with their peers and feel understood: "*it´s about the sense of feeling understood in every way, when angry or sad, asking the question 'why'?'. . . the fact that I have to adapt my life to this new situation*" (female leader, 5.41); and "*there are many things you don´t talk about at home because you are afraid that this fear inside you may pass on to other people, the doubts you have, everything, then emotionally, when you talk to people who are in the same situation to you, or have been there before . . . each addresses it in a different way, but they are peers*" (female leader, 5.55).

## A more positive view of their experience, helping them become active in self-care and empowered in the self-management of their condition

Participants reported that the peer education program had helped them reflect on their situation, "*what I´ve experienced here has made me think*" (female survivor, 4.103), feel listened to and understood as well as improve their communication skills "*they help you live with others*" (female survivor, 3.86), have a more positive view of their experience, "*we shared our things. We´ve brought out our emotions; we´ve cried but we´ve also laughed*" (female survivor, 3.10), and develop resilience "*we have to keep fighting*" (female survivor, 4.65); "*it improves the attitude that you may have towards your situation, your condition*" (female survivor, 4.81); and "*in this sense, it has made me stronger as a person at the individual level . . . regarding the disease, it has strengthened me, I mean, [it makes me] think that I´m very lucky*" (female survivor, 4.104). In turn, this helped to widen their social support network and encouraged them to participate more actively in their self-care and the self-management of their condition: "*the approach of the course is such that you become aware of your condition and look after yourself*" (female survivor, 3.110); and "*it´s about rethinking how we should behave in relation to the disease so that we can manage it*" (female survivor, 4.115). This is achieved by putting into practice the methods learned in the program for setting small goals for change or improvement: "*the bit about commitment was useful . . . because you committed yourself and you did it . . . you set yourself a goal and went for it, you tried to do it for the following week*" (female survivor, 2.48); and "*with that sense of incapacity, suddenly you realize that you can set yourself a goal . . . that you are able to reach or do it. . . for me, this is a tool you can hold on to so that you don´t drown in your sorrow*" (female survivor, 3.72).

Peer leaders corroborated the contribution of the education program to participants' attitude to change through an approach based on setting small goals: "*in the workshop, we learn to set small goals so that we can make these small changes and very simple objectives, because otherwise we get unstuck along the way*" (female leader, 5.68); and "*the workshop helps us to set goals, to make small changes and especially to become aware of where we are now, how we are doing, how well are we eating, how we are living, what emotions we are feeling, being more aware, and then each of us starts making the changes we think we need to make*" (female leader, 5.69). Peer leaders also highlighted that the program helps cancer survivors to improve their communication skills, "*cancer patients turn in on themselves, and in this way, they start communicating more and open up, I think they open up a lot . . . I think it does them good; they leave very happy*" (male leader, 5.2), and develop problem-solving skills, "*these new strategies you learn,*

*for conflict resolution, are very useful*" (female leader, 5.63). Additionally, they affirmed that all of this leads to participants becoming more actively involved in self-care and the self-management of their condition, "*improving in terms of exercise, diet, quality of life. . .*" (female leader, 5.46), which helps them to start to empower themselves in the management of their current situation, "*these small challenges we set each week. . . I think we finish the workshop having learnt a lot, with a lot of information, with a lot of understanding, that we felt, well, understood and appreciated and above all really empowered*" (female leader, 5.58); and "*they are now active patients, they are playing their part*" (female leader, 5.79).

## Discussion

The results of this study indicate that the participants endorse the value of the "Looking after my health after cancer" peer-led active patient education program as an intervention to increase awareness of their perceived unmet needs and respond to them, as well as promote the acquisition of skills and tools for activation in self-care and self-management of their own health to improve their quality of life. These findings provide evidence that this education program has a positive impact on cancer survivors and their close relatives, considering their satisfaction with the program and the positive learning experience they describe after their participation. Notably, both survivors and peer leaders indicated that the peer-led nature of the program enables participants to open up, share their life experiences, and feel more understood. Based on this, the researchers speculate that, among other factors, the program being peer led, that is, mediated by people who have experienced a similar situation and have received training to become leaders and run the sessions, may have contributed to its success. According to social comparison theory, people with acute or chronic illnesses use comparisons with others who have gone through similar experiences to cope with their situation, reduce the perceived threat and find ways to address challenges [29]. Peers are perceived as models who have had a similar experience before them and who have come through it. Hence, meeting peers offers an opportunity to make a positive comparison, in that they get the chance to get to know other people who have experienced and overcome similar stressful situations [30]. Previous studies provide evidence of the positive results achieved with expert patients such as leaders in peer education programs in the case of people with chronic illnesses [31–33]. Recently, along similar lines to this study, there have been other reports of programs for training cancer survivors, for them to become leaders in peer education programs, such as the Peer Connect program, based on the train-the-trainer model with breast cancer survivors [34], and peer navigators in the TrueNTH Peer Navigation Training Program, with prostate cancer survivors and their caregivers [35].

In this study, the researchers have also observed that participation in a peer-led program enables cancer survivors and their family caregivers to become more aware of late adverse effects and their own unmet felt needs, at physical, emotional, social, and spiritual levels, and in turn, feel better understood concerning all aspects of what is involved in living the "new normal" after cancer. The group-based nature of the program enables participants to share their experiences and find mutual understanding and support, which in turn serve as motivational stimuli ("If they can, so can I") to cope with everything involved in living in survivorship. Thanks to this, cancer survivors can view their experiences more positively, leading to greater self-care activation and empowerment for self-management of their condition. In line with this, though still relatively few, some other studies with cancer survivors have also used group dynamics techniques with positive results, including a randomized controlled trial in breast cancer survivors who participated in support-expression discussion groups that reported significant reductions in loneliness, increases in hopefulness, and improvements in

quality of life [36]. Further, other education programs for cancer survivors led by peers are emerging. The researchers identified four peer education programs that have shown positive results consistent with those observed in this study. Based on the *Chronic Disease Self-Management Program* [21], the same model as that used for this "Looking after my health after cancer" peer-led active patient education program, Risendal et al. [37] confirmed the viability and acceptance of a self-care program focused around peer learning in cancer survivors. The *Breast Health Buddy* program for breast cancer survivors also achieved good results, participants reporting greater access to and use of alternative support sources, greater ability to cope with the stress of their situation, and better quality of life and adjustment than survivors who received usual care [38]. A peer learning program in a Norwegian cancer hospital for people with cancer, including cancer survivors, was found to have positive effects, peers helping to sustain hope and provide ways of coping, share experiences and offer support complementary to that received from family and health professionals [39]. Lastly, the *Nuevo Amanecer* (New Dawn) program for breast cancer survivors also found that participants reported improvements in their skills for managing stress and mood changes, participated more in their own care, and felt a greater sense of hope and trust in their peers [40]. These types of educational interventions to promote self-management are able to improve the management of adverse effects, mood, and quality of life [41]. Further, they allow participants to obtain emotional support and information from their peers openly and without prejudgment, empower themselves by improving their knowledge and skills, regain their confidence and control, cope with the situation, return to goal setting, and adopt a positive attitude [42].

## Strengths and limitations

A key strength of this study was the inclusion of a focus group with peer leaders that enabled the researchers to make comparisons with the perception of cancer survivors and caregivers participating in the program, triangulate and enrich the findings and confirm the positive rating and impact of the program. In relation to this, it has been reported that peer support programs may also be useful and a positive experience for peer leaders themselves, who report emotional gains, with greater self-confidence, a sense of pride, and enhanced self-esteem, as well as a greater sense of connection with others [43]. Secondly, the qualitative perspective allowed for an in-depth exploration of the participants' and peer leaders' experiences and views, providing a more comprehensive insight into the questions of interest in this study.

This study has certain limitations. On the one hand, only two caregivers participated in the discussion groups for participants, limiting the knowledge of caregivers' views concerning the usefulness and impact of the program on the close relatives of cancer survivors. Another limitation is the low participation of men in this peer-led active patient education program and consequently also in the study. This pattern is also evident in the literature on the Stanford's Chronic Disease Self-Management Education program (on which this program is based), which shows that despite the availability and benefits of the program, male participation remains low [44]. Further, the researchers should highlight the need to complement this study with quantitative research that would allow to assess the level of satisfaction with the program and its impact on quality of life and level of self-care activation in more detail. It is also important to undertake a longitudinal study that would enable to evaluate what participants have learned during and after the education program to assess its impact in the short and long term.

## Conclusion

The "Looking after my health after cancer" peer-led active patient education program has shown a good impact on cancer survivors and their close relatives in terms of satisfaction with

the program and a positive learning experience. Participants were able to share their experiences and feel understood by their peers in the "new normal" of survivorship, in turn, this seemed to help them to become more aware of and better address their unmet felt needs, leading to a more positive view of their experiences and greater self-care activation and better self-management of their condition.

This study provides evidence of the benefits of a new type of intervention based on peer learning in cancer survivors. Nonetheless, given the current challenge of providing integrated care and greater follow-up for this population and the need to strengthen the evidence regarding the benefits and the impact of this type of intervention, continued efforts to design, implement and evaluate interventions such as the one the considered in this study are warranted, seeking to reach greater consensus on the most suitable way to address the unmet felt needs of cancer survivors with the goal of improving their quality of life.

## Acknowledgments

The researchers are grateful to all the participants and peer leaders who have made this program possible since its inception. Further, they would like to thank the cancer survivors, caregivers, and peer leaders who participated in this study for their contribution. It would not have been able to conduct this study without their support.

## Author Contributions

**Conceptualization:** Ainhoa Ulibarri-Ochoa, Sheila Sánchez-Gómez, Estíbaliz Gamboa-Moreno, Lourdes Ochoa de Retana-García.

**Data curation:** Ainhoa Ulibarri-Ochoa, Estíbaliz Gamboa-Moreno, Begoña Belarra-Tellechea, Lourdes Ochoa de Retana-García.

**Formal analysis:** Ainhoa Ulibarri-Ochoa, Sheila Sánchez-Gómez.

**Funding acquisition:** Estíbaliz Gamboa-Moreno, Lourdes Ochoa de Retana-García.

**Investigation:** Ainhoa Ulibarri-Ochoa, Estíbaliz Gamboa-Moreno, Begoña Belarra-Tellechea, Lourdes Ochoa de Retana-García.

**Methodology:** Ainhoa Ulibarri-Ochoa, Sheila Sánchez-Gómez.

**Project administration:** Estíbaliz Gamboa-Moreno, Lourdes Ochoa de Retana-García.

**Resources:** Estíbaliz Gamboa-Moreno, Lourdes Ochoa de Retana-García.

**Software:** Ainhoa Ulibarri-Ochoa, Sheila Sánchez-Gómez.

**Supervision:** Ainhoa Ulibarri-Ochoa, Sheila Sánchez-Gómez.

**Validation:** Ainhoa Ulibarri-Ochoa, Sheila Sánchez-Gómez, Estíbaliz Gamboa-Moreno, Irene Duo-Trecet, Lucia Garate-Echenique, Begoña Belarra-Tellechea, Lourdes Ochoa de Retana-García.

**Visualization:** Ainhoa Ulibarri-Ochoa.

**Writing – original draft:** Ainhoa Ulibarri-Ochoa, Sheila Sánchez-Gómez.

**Writing – review & editing:** Ainhoa Ulibarri-Ochoa, Sheila Sánchez-Gómez, Estíbaliz Gamboa-Moreno, Irene Duo-Trecet, Lucia Garate-Echenique, Begoña Belarra-Tellechea, Lourdes Ochoa de Retana-García.

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
