## [Decision Letter · Decision Letter 0]

28 Nov 2022

PONE-D-22-19858Impact of the “Looking after my health after cancer” peer-led active patient education program on cancer survivors and their caregivers: A qualitative studyPLOS ONE

Dear Dr. Ulibarri-Ochoa,

Thank you for submitting your manuscript to PLOS ONE. After careful consideration, we feel that it has merit but does not fully meet PLOS ONE’s publication criteria as it currently stands. Therefore, we invite you to submit a revised version of the manuscript that addresses the points raised during the review process.

We look forward to receiving your revised manuscript.

Kind regards,

Sandra Boatemaa Kushitor, Ph.D.

Academic Editor

PLOS ONE

https://journals.plos.org/plosone/s/fileid=ba62/PLOSOne_formatting_sample_title_authors_affiliations.pdf.

“The study was awarded the III Research Prize for Nursing in Cancer and Palliative Care from the Angel Muriel Foundation. Further, a grant to support nursing research was received from the College of Registered Nurses of Gipuzkoa (COEGI 2019). Likewise, with funding from Biodonostia Health Research Institute, the manuscript has been translated and reviewed by the scientific editors of Ideas Need Communicating Language Services.”

:

Additional Editor Comments:

Dear Authors,

Congratulations for writing this manuscript. Please revise it based on the minor comments suggested by the reviewer and I.

Reviewers' comments:

Reviewer's Responses to Questions

**Comments to the Author**

1. Is the manuscript technically sound, and do the data support the conclusions?

Reviewer #1: Yes

2. Has the statistical analysis been performed appropriately and rigorously? 

Reviewer #1: N/A

3. Have the authors made all data underlying the findings in their manuscript fully available?

Reviewer #1: Yes

4. Is the manuscript presented in an intelligible fashion and written in standard English?

Reviewer #1: Yes

5. Review Comments to the Author

Reviewer #1: PONE-D-22-19858 comments

The authors conducted research that touches on a very important topic. They employed a phenomenology approach to the design.

The objectives of the study are stated as exploring: how cancer survivors and their caregivers rate the experience of participating in the program, the extent to which the program helps them understand and address their unmet felt needs, and the extent to which it helps them improve their activation for self-care and self-management.

Methods

Overall, the methods section is well outlined with details. I would however have liked to see the methods presented using the five steps/stages of the phenomenology design used, as outlined below:

1. Bracketing and phenomenological reduction.

2. Delineating units of meaning.

3. Clustering of units of meaning to form themes.

4. Summarising each interview, validating it and where necessary modifying it.

5. Extracting general and unique themes from all the interviews and making a composite summary.

Line 178 ‘has’ should be ‘had’

Results.

Table 2

It is not obvious what the letter 'M' in the second column represents. Is it mean or median?

Secondly, this value should be a measure of central tendency and its appropriate measure of dispersion in parentheses e.g., mean (sd), median (iqr)

The ratio of Females to Males is quite large. Was this part of the design or a flaw in sampling? Was it not necessary to have equal gender representation? This is especially important because you mention that you aimed at achieving a heterogeneous sample.

In my opinion, it would have been better to see the results presented per objective. The way they have been presented leaves one wondering if there was a reflection on the individual objectives of the study in a way that would enable the reader to understand if the study questions were answered or not. I would suggest that the results are reorganised per the objectives.

Line 255. “Satisfaction with the program and the positive learning experience.” Could these be two independent themes? There are several areas where the results are presented this way. I would imagine that the main themes would be meaningful enough to stand on their own. Either the wrong title of themes was used, or more than two or more themes are being combined under one heading in a way that makes them seem different.

Line 416, 417. “Having a more positive view of their experience and becoming active in self-care and more empowered in self-management of their condition.” Are these not two independent themes? Being more empowered in self-management (self-efficacy) is different from having a positive view (self-image)

There are numerous areas in the results section where only positive findings were reported. I might have missed this, but it is rare for any study of this kind not to have a single negative or surprising or unexpected finding. I am not doubting the results as presented. I am simply wondering if the authors took time to either ask about things that didn’t go well or any findings that might have not been positive. These too are important to report about, if they exist.

Discussion

When researchers use comparative methods, they organise groups of findings with the objective of developing new theories about phenomena. It is not obvious what new theory about interventions of this kind was discerned from the process. What is the overriding communication finding(s) from the study? This is not well articulated.

Line 458-459 “The success of this program is attributed to the fact that it is peer led, that is, mediated by people who have experienced a similar situation and have received training to become leaders and run the sessions.”

For most interventions of this kind, their effects cannot be attributed to one factor. Moreover, the study was not designed to show cause and effect. it is therefore not obvious that the fact that it was per led is the only reason it was successful. This was not a process evaluation study. I am, therefore, not sure you had all the variables to explain why the intervention worked the way it did. At best, the authors can only hypothesize about the factors that could have contributed to its success.

Strengths and limitations section:

One potential limitation is the use of one method of data collection – focus groups. Studies of this kind could benefit from individual interviews and focus groups unless the goal is to achieve consensus, which was not the case in this study.

Line 506: triangulation

Triangulation is often used in dealing with different groups and different research methods. It is not clear how triangulation was done in this study. It would be great if the authors described this in detail.

Reflexivity

It is good practice to reflect on the authors’ own biases in designing and implementing qualitative studies. For this reason, authors of qualitative studies, especially interpretive designs such as the one used in this study are advised to reflect on how their experiences might have influenced the conduct of the study. The authors are therefore advised to include a note on reflexivity. How did the researchers' training and life and professional experiences influence the study design, and data collection, interpretation and reporting?

6. PLOS authors have the option to publish the peer review history of their article (what does this mean?). If published, this will include your full peer review and any attached files.

Reviewer #1: No

---

## [Author Response · Author response to Decision Letter 0]

26 Dec 2022

December 26th, 2022 

Dear academic editor and reviewers,

Manuscript PONE-D-22-19858, entitled ‘Impact of the “Looking after my health after cancer” peer-led active patient education program on cancer survivors and their caregivers: A qualitative study’:

We are grateful for the comments provided. These comments have undoubtedly helped us to improve our manuscript.

Please find enclosed the response to reviewers and the completely revised version of the manuscript (PONE-D-22-19858) in which we have addressed all the comments of the reviewers and implemented their suggestions. All changes are shown using the Track changes tool and are also highlighted in yellow. All co-authors have approved these changes. The changes in the wording have been reviewed by scientific editors specialized in health sciences articles. 

Below are listed the comments and suggestions of the academic editor and reviewers and the response of the authors:

Academic Editor´s Comments

 Sandra Boatemaa Kushitor, Ph.D.

1. Thank you for submitting your manuscript to PLOS ONE. After careful consideration, we feel that it has merit but does not fully meet PLOS ONE’s publication criteria as it currently stands. Therefore, we invite you to submit a revised version of the manuscript that addresses the points raised during the review process.

Authors' response: 

We appreciate opportunity to revise our manuscript. We are grateful for the privilege of having our manuscript reviewed and for the valuable suggestions of the reviewers which we believe have helped us to improve it. 

We have implemented changes in accordance with the reviewers' suggestions and these are shown using the Track changes tool and also highlighted in yellow.

The changes in the wording have been reviewed by scientific editors specialized in health sciences articles.

https://journals.plos.org/plosone/s/fileid=ba62/PLOSOne_formatting_sample_title_authors_affiliations.pdf.

Authors' response: 

We have addressed the style requirements for the PLOS One journal, including those for file naming and style templates. 

3. 

“The study was awarded the III Research Prize for Nursing in Cancer and Palliative Care from the Angel Muriel Foundation. Further, a grant to support nursing research was received from the College of Registered Nurses of Gipuzkoa (COEGI 2019). Likewise, with funding from Biodonostia Health Research Institute, the manuscript has been translated and reviewed by the scientific editors of Ideas Need Communicating Language Services.”

Authors' response:

The funders played no role in the design or conduct of the study. 

We have added the statement that "The funders had no role in study design, data collection and analysis, decision to publish, or preparation of the manuscript". 

This sentence is highlighted in yellow in the revised cover letter. Thank you very much for changing the online submission form on our behalf.

Authors' response:

We have checked all the references.

To our knowledge, none of the papers cited in the manuscript have been retracted. On the other hand, in accordance with the reviewers' suggestions for changes, we have added some text, and this is supported by a new reference (line 239, reference number 28) which has been included in the reference list of the manuscript (from line 695 to line 696): Groenewald T. A Phenomenological Research Design Illustrated. Int J Qual Methods. 2004; 3(1): 42-55. doi: 10.1177/160940690400300104.

In addition, to explain the limitation of the lower participation of men (see paragraph from line 589 to line 593 in the subsection of "Strengths and limitations"), we have included reference number 44 (from line 742 to line 744): Lee M, Bergeron CD, Ahn S, Towne SD, Mingo CA, Robinson KT et al. Engaging the Underrepresented Sex: Male Participation in Chronic Disease Self-Management Education (CDSME) Programs. AM J Mens Health. 2018; 12(4): 935-943. doi: 10.1177/1557988317750943.

Reviewer 1

5. Dear Authors,

Congratulations for writing this manuscript. Please revise it based on the minor comments suggested by the reviewer and I.

Authors' response:

Thank you for your careful review. We are grateful for your feedback and your important reflections that we believe have helped us to improve the manuscript.

6. The authors conducted research that touches on a very important topic. They employed a phenomenology approach to the design.

The objectives of the study are stated as exploring how cancer survivors and their caregivers rate the experience of participating in the program, the extent to which the program helps them understand and address their unmet felt needs, and the extent to which it helps them improve their activation for self-care and self-management.

Authors' response:

Thank you for recognizing the importance of the central theme of our research.

7. Methods:

Overall, the methods section is well outlined with details. I would however have liked to see the methods presented using the five steps/stages of the phenomenology design used, as outlined below:

1. Bracketing and phenomenological reduction.

2. Delineating units of meaning.

3. Clustering of units of meaning to form themes.

4. Summarising each interview, validating it and where necessary modifying it.

5. Extracting general and unique themes from all the interviews and making a composite summary.

Authors' response:

We agree with the suggestion to make the five steps of the phenomenological design more explicit. In the Data analysis subsection of Materials and methods, we have expanded the description of the analysis following these five recommended steps. Specifically, we have included the following paragraph (from line 238 to line 255):

“These data were analyzed using phenomenological methods following a simplified version of Hycner’s process [28] which has five steps/stages: 1) Bracketing and phenomenological reduction: with an attitude of openness to any meaning that emerged, two researchers repeatedly read the information provided by the key informants in focus groups to become familiar with the testimonies, identify units of meaning and develop a holistic sense of what was expressed by the survivors, caregivers, and peer leaders. Each focus group was identified by a numerical code (from 1 to 5), and in turn, each testimony in each focus group was also coded; 2) Delineating units of meaning: units of relevant meaning were extracted and reviewed, any redundant units being eliminated; 3) Clustering of units of meaning to form themes: the units of meaning were grouped to form themes reflecting the participants’ experiences; 4) Summarizing of each focus group, validating the themes and where necessary modifying them: a summary was produced containing all the themes generated from the units of meaning. A validity check was then performed, going back to the informants’ testimonies to assess whether the essence of the testimonies in the focus groups had been captured correctly and modifying the summary as appropriate; and 5) Extracting general and unique themes from all the focus groups and making a composite summary: themes common to most of the focus group testimonies were identified, as well as individual variations, and the analysis was concluded by writing a composite summary that reflected the transformation of the everyday expressions of the study participants into expressions appropriate to the scientific discourse supporting the objectives of this research”.

8. Methods:

Line 178 ‘has’ should be ‘had’

Authors' response:

Now in line 192, "had" has been changed to "has".

9. Results:

Table 2. It is not obvious what the letter 'M' in the second column represents. Is it mean or median?

Secondly, this value should be a measure of central tendency and its appropriate measure of dispersion in parentheses e.g., mean (sd), median (iqr)

Authors' response:

The letter 'M' in the second table refers to the mean. Standard deviation has been included in parentheses M (SD) = 59.13 (11.16).

10. Results:

The ratio of Females to Males is quite large. Was this part of the design or a flaw in sampling? Was it not necessary to have equal gender representation? This is especially important because you mention that you aimed at achieving a heterogeneous sample.

Authors' response:

Indeed, the initial objective was to achieve a heterogeneous sample. This was an ambitious aspiration, however, given that many more women than men participate in the "Looking after my health after cancer" education program, with a female-to-male ratio among participants of 4:1, and the sample was obtained by theoretical intentional sampling. For these reasons, more women than men finally participated. 

In relation to this sense, there is evidence that shows that females are more likely than males to participate in evidence-based health promotion and disease prevention programs targeting middle-aged and older adults. In particular, the current scientific literature on Stanford’s Chronic Disease Self-Management Education programs, on which our research study is based, states that despite the availability and benefits of these programs, male participation remains low. 

Source of information [44]: Lee M, Bergeron CD, Ahn S, Towne SD, Mingo CA, Robinson KT et al. Engaging the Underrepresented Sex: Male Participation in Chronic Disease Self-Management Education (CDSME) Programs. AM J Mens Health. 2018; 12(4): 935-943. doi: 10.1177/1557988317750943

We recognize this imbalance to be a limitation of the study that should be addressed in future studies and it has been included, and duly explained, in the “Strengths and limitations” subsection (from line 589 to line 593).

11. Results:

In my opinion, it would have been better to see the results presented per objective. The way they have been presented leaves one wondering if there was a reflection on the individual objectives of the study in a way that would enable the reader to understand if the study questions were answered or not. I would suggest that the results are reorganised per the objectives.

Authors' response:

Thanks for this suggestion. Seeking to make it easier for readers to understand which study questions have been addressed, we have now explained - in the second subsection of Results- that in response to the three objectives of the study, five main themes emerged from the content analysis of the data. Further, we have specified which themes correspond to each objective, and for clarity, have added a new table (Table 3) showing the link between each of the objectives and these five main themes (from line 298 to line 308).

12. Results:

Line 255. “Satisfaction with the program and the positive learning experience.” Could these be two independent themes? There are several areas where the results are presented this way. I would imagine that the main themes would be meaningful enough to stand on their own. Either the wrong title of themes was used, or more than two or more themes are being combined under one heading in a way that makes them seem different.

Authors' response:

We have related satisfaction with the program and the positive learning experience because these two units of meaning are closely related. Program participants are very satisfied with their participation in that their participation has been a learning experience that they value very positively. To clarify this, the title of this theme has been modified to: “Satisfaction with the program as a positive learning experience” (line 313).

13. Results:

Line 416, 417. “Having a more positive view of their experience and becoming active in self-care and more empowered in self-management of their condition.” Are these not two independent themes? Being more empowered in self-management (self-efficacy) is different from having a positive view (self-image).

Authors' response:

The concept of having a more positive view does not refer to their self-image, but rather to the ability to face the new survival situation in a more positive way, with a more positive attitude. The fact that they see the reality of the new survival phase in a more positive way thanks to the support of their peers helps them to motivate themselves and to participate more actively in their self-care and self-management of their process. That is why we have related these two units of meaning. To clarify this, the title of this theme has been modified to: “A more positive view of their experience, helping them become active in self-care and empowered in the self-management of their condition” (from line 485 to line 487). 

14. Results:

There are numerous areas in the results section where only positive findings were reported. I might have missed this, but it is rare for any study of this kind not to have a single negative or surprising or unexpected finding. I am not doubting the results as presented. I am simply wondering if the authors took time to either ask about things that didn’t go well or any findings that might have not been positive. These too are important to report about, if they exist.

Authors' response:

Thank you very much for your appreciation and for giving us the opportunity to explain this aspect. As can be seen in the semi-structured guide consisting of open-ended questions, there are several questions aimed at identifying areas for improvement such as: Was it what you expected? To what extent has it met your expectations at the outset? Was there anything you felt was missing? What would you change? Did you experience any difficulties participating in the program? Are there any issues related to your experience with this program that you think we should know that we have not yet touched on, that you consider important and that you would like to add?

However, the participants made a very positive assessment of their participation in the program and mainly highlighted the positive aspects. Their only concern was that they felt it was important to consider enhancing preparedness prior to their participation in the program, since they report being afraid of reliving difficult situations from the near past during the diagnosis and treatment phase.

The positive experience and results of participation in the program were also corroborated by the focus group carried out with the program peer leaders.

15. Discussion:

When researchers use comparative methods, they organise groups of findings with the objective of developing new theories about phenomena. It is not obvious what new theory about interventions of this kind was discerned from the process. What is the overriding communication finding(s) from the study? This is not well articulated.

Authors' response:

We have sought to clarify the key study findings in the Discussion. In particular, at the beginning of this section, a sentence has been added that summarizes and emphasizes the main findings of this study, and with this, we also seek to ensure that the objectives are better linked to the results and the main findings (from line 525 to line 531):

“The results of this study indicate that the participants endorse the value of the “Looking after my health after cancer” peer-led active patient education program as an intervention to increase awareness of their perceived unmet needs and respond to them, as well as promote the acquisition of skills and tools for activation in self-care and self-management of their own health to improve their quality of life. Our findings provide evidence that this education program has a positive impact on cancer survivors and their close relatives, considering their satisfaction with the program and the positive learning experience they describe after their participation.”

16. Discussion:

Line 458-459 “The success of this program is attributed to the fact that it is peer led, that is, mediated by people who have experienced a similar situation and have received training to become leaders and run the sessions.”

For most interventions of this kind, their effects cannot be attributed to one factor. Moreover, the study was not designed to show cause and effect. It is therefore not obvious that the fact that it was per led is the only reason it was successful. This was not a process evaluation study. I am, therefore, not sure you had all the variables to explain why the intervention worked the way it did. At best, the authors can only hypothesize about the factors that could have contributed to its success.

Authors' response:

Thanks for this comment. We fully agree, and therefore, from line 531 to line 535, we have modified the wording to: “ Notably, both survivors and peer leaders indicated that the peer-led nature of the program enables participants to open up, share their life experiences and feel more understood. Based on this, we speculate that, among other factors, the program being peer led, that is, mediated by people who have experienced a similar situation and have received training to become leaders and run the sessions, may have contributed to its success”.

17. Strengths and limitations section:

One potential limitation is the use of one method of data collection – focus groups. Studies of this kind could benefit from individual interviews and focus groups unless the goal is to achieve consensus, which was not the case in this study.

Authors' response:

We understand this statement and although it is true that we could have also carried out in-depth individual interviews, we opted for focus groups as the main strategy for data generation since they offer the possibility of interaction between peers (people who have gone through the same experience) and this can facilitate obtaining more detailed information about the collective experience of participating in the program.

18. Strengths and limitations section:

Line 506: triangulation

Triangulation is often used in dealing with different groups and different research methods. It is not clear how triangulation was done in this study. It would be great if the authors described this in detail.

Authors' response:

Thanks for the recommendation. We have sought to better explain how the triangulation process worked and why we decided to carry out a fifth focus group with program peer leaders. For this, we have included the following sentence in the Rigor criteria subsection of Materials and methods (from line 270 to line 276):

“Given that the peer leaders had also previously participated as cancer survivors or caregivers in the program and had subsequently been rigorously trained to run the sessions of this program, we believed it important to complement the perspective of participants with that of the peer leaders. In particular, they were the people who oversaw the program and could potentially provide relevant information about what happened in the sessions and whether what was expressed by the participants agreed with their impressions as peer leaders, and thus be able to compare, corroborate and enrich the findings”.

19. Reflexivity

It is good practice to reflect on the authors’ own biases in designing and implementing qualitative studies. For this reason, authors of qualitative studies, especially interpretive designs such as the one used in this study are advised to reflect on how their experiences might have influenced the conduct of the study. The authors are therefore advised to include a note on reflexivity. How did the researchers' training and life and professional experiences influence the study design, and data collection, interpretation and reporting?

Authors' response:

In the Data analysis subsection of Material and methods, a paragraph on reflexivity has been included, explaining how researchers' training and life and professional experiences influence the data analysis (from line 256 to line 262):

“Regarding the concept of reflective practice, it is important to point out that one of the two researchers who carried out the initial thematic analysis had first-hand experience of the reality of cancer survivorship as a professional and as a caregiver. This experience helped to enrich the reflective and analytical process and the interpretation of the data that inform the research question. It is also important to note that the two researchers who performed the analysis were trained in qualitative methodology. Further, the other researchers who corroborated the analysis were nurses who were familiar with the education program and the reality of cancer survivors and caregivers”.

Reviewer 2

20. Line 86.

Different font

Authors' response:

In line 86, the word "early" was written in a different font size. This has been amended to the same font size as the rest of the text: Times New Roman 10.

21. Line 163. 

The authors should provide more information about how the content of the material was designed. Who was involved in the design and any validity measures that were used.

Authors' response:

We have added an explanation of the process of designing, developing and validating the content of the educational program. We have also explained who was involved in the design and piloting of the program (from line 176 to line 189):

“The content of the program was designed taking the structure of Stanford's Chronic Disease Self-Management Education Program as a reference. To contextualize it to the care of cancer survivors, a literature review was conducted to identify potential felt unmet needs in this population. A key study was the Spanish Group for Cancer Patients (GEPAC)’s Report on the needs of cancer survivors based on a study conducted in Spain a few years earlier. The content of the materials for the various sessions of the program was developed by a group composed of two oncologists, a psychiatrist, a psychologist, two oncology nurses, a nurse responsible for the Patient Care Service, a management nurse, and two primary care doctors and three primary care nurses trained in the program's methodology. The materials developed by this group were reviewed by three patients (who were to be program peer leaders) and two patient associations (Spanish Association Against Cancer and Association of Women Affected by Breast and Gynecological Cancer of Gipuzkoa). With their contributions, various materials were reworked. Finally, the program was piloted in a group of seven cancer survivors and feedback was requested after each session. Having completed this first pilot of the program, the content manual was finalized, considering all the comments and changes suggested, and the program was validated”.

22. Line 192. 

If the moderator is part of the authors, would you to mention the persons´ name.

Authors' response:

The authors who in turn were peer leaders of the focus groups of the research study have been named with their initials in the “Data collection” section (line 206).

23. Line 220.

Information about how themes were generated are missing. Provide a description of how inductive and deductive codes were derived.

Authors' response:

As previously explained in our response to a recommendation from Reviewer 1, in the Data analysis subsection of Material and methods, a paragraph has been included explaining how the data were analyzed and also how the different testimonies were coded (from line 238 to line 255):

“These data were analyzed using phenomenological methods following a simplified version of Hycner’s process [28] which has five steps/stages: 1) Bracketing and phenomenological reduction: with an attitude of openness to any meaning that emerged, two researchers repeatedly read the information provided by the key informants in focus groups to become familiar with the testimonies, identify units of meaning and develop a holistic sense of what was expressed by the survivors, caregivers, and peer leaders. Each focus group was identified by a numerical code (from 1 to 5), and in turn, each testimony in each focus group was also coded; 2) Delineating units of meaning: units of relevant meaning were extracted and reviewed, any redundant units being eliminated; 3) Clustering of units of meaning to form themes: the units of meaning were grouped to form themes reflecting the participants’ experiences; 4) Summarizing of each focus group, validating the themes and where necessary modifying them: a summary was produced containing all the themes generated from the units of meaning. A validity check was then performed, going back to the informants’ testimonies to assess whether the essence of the testimonies in the focus groups had been captured correctly and modifying the summary as appropriate; and 5) Extracting general and unique themes from all the focus groups and making a composite summary: themes common to most of the focus group testimonies were identified, as well as individual variations, and the analysis was concluded by writing a composite summary that reflected the transformation of the everyday expressions of the study participants into expressions appropriate to the scientific discourse supporting the objectives of this research”.

24. Line 223.

Expand on this more. Indicate the role of each of these researchers and at what stage they were involved in the analysis.

Authors' response:

In the Rigor Criteria subsection of Material and Methods, the role of each of the researchers and at what stage they were involved in the analysis has been detailed (from line 265 to line 268).

25. Line 269. 

The numbering for this is different from the others.

Authors' response:

The citation code 5.137 follows the same format as the rest of the citations: the first number refers to the fifth focus group of peer leaders and the number 137 to the citation number (line 327).

26. Line 333.

The quotes are many. It will be good to present the types of physical challenges they experience. 

Authors' response:

In response to this comment, the paragraph concerning the physical needs expressed by cancer survivors has been reworked (from line 386 to line 396), seeking to make it clearer what physical challenges they face.

27. Fig 1.

It will be good to find a picture with sharper resolution. The image is difficult to read.

Authors' response:

Thank you very much for the indications made to improve the quality of the figures. To improve the resolution of Figures 1 and 2, the Preflight Analysis and Conversion Engine (PACE) digital diagnostic tool has been used (https://pacev2.apexcovantage.com/)

Our very best, 

The authors

---

## [Editor Report · Decision Letter 1]

13 Jan 2023

PONE-D-22-19858R1Impact of the “Looking after my health after cancer” peer-led active patient education program on cancer survivors and their caregivers: A qualitative studyPLOS ONE

Dear Dr. Ulibarri-Ochoa,

Thank you for submitting your manuscript to PLOS ONE. After careful consideration, we feel that it has merit but does not fully meet PLOS ONE’s publication criteria as it currently stands. Therefore, we invite you to submit a revised version of the manuscript that addresses the points raised during the review process. Kindly edit your manuscript and remove all first person words. Avoid I and we, use phrases such as 'the researcher', the authors  ectera.  

We look forward to receiving your revised manuscript.

Kind regards,

Sandra Boatemaa Kushitor, Ph.D.

Academic Editor

PLOS ONE
---

## [Author Response · Author response to Decision Letter 1]

31 Jan 2023

January 30th, 2023

Editor-in-Chief, PLOS One

Dear Dr Chenette,

Manuscript PONE-D-22-19858R1, entitled ‘Impact of the “Looking after my health after cancer” peer-led active patient education program on cancer survivors and their caregivers: A qualitative study’

We sincerely appreciate the interest of the PLOS One journal in our work, and we are grateful for the comments provided. These comments have undoubtedly helped us to improve our manuscript.

Please find enclosed a revised version of the manuscript (PONE-D-22-19858R1) in which we have addressed all the comments of the Academic Editor and implemented her suggestions. All changes are shown using the Track changes tool and are also highlighted in yellow. All co-authors have approved these changes. 

Thank you again for the privilege of having our work reviewed, and we hope that you find our responses satisfactory. 

 We look forward to your final decision regarding our manuscript.

Our very best, 

The authors.

---

## [Decision Letter · Decision Letter 2]

7 Feb 2023

Impact of the “Looking after my health after cancer” peer-led active patient education program on cancer survivors and their caregivers: A qualitative study

PONE-D-22-19858R2

Dear Dr. Ulibarri-Ochoa,

We’re pleased to inform you that your manuscript has been judged scientifically suitable for publication and will be formally accepted for publication once it meets all outstanding technical requirements.

Kind regards,

Nabeel Al-Yateem, PhD

Academic Editor

PLOS ONE

Additional Editor Comments (optional):

Reviewers' comments:

Reviewer's Responses to Questions

**Comments to the Author**

1. If the authors have adequately addressed your comments raised in a previous round of review and you feel that this manuscript is now acceptable for publication, you may indicate that here to bypass the “Comments to the Author” section, enter your conflict of interest statement in the “Confidential to Editor” section, and submit your "Accept" recommendation.

Reviewer #1: All comments have been addressed

2. Is the manuscript technically sound, and do the data support the conclusions?

Reviewer #1: Yes

3. Has the statistical analysis been performed appropriately and rigorously? 

Reviewer #1: Yes

4. Have the authors made all data underlying the findings in their manuscript fully available?

Reviewer #1: Yes

5. Is the manuscript presented in an intelligible fashion and written in standard English?

Reviewer #1: Yes

6. Review Comments to the Author

Reviewer #1: (No Response)

7. PLOS authors have the option to publish the peer review history of their article (what does this mean?). If published, this will include your full peer review and any attached files.

Reviewer #1: **Yes: **Daniel Semakula

---

## [Editor Report · Acceptance letter]

14 Feb 2023

PONE-D-22-19858R2 

Impact of the “Looking after my health after cancer” peer-led active patient education program on cancer survivors and their caregivers: A qualitative study 

Dear Dr. Ulibarri-Ochoa:

I'm pleased to inform you that your manuscript has been deemed suitable for publication in PLOS ONE. Congratulations! Your manuscript is now with our production department. 

Kind regards, 

on behalf of

Dr. Nabeel Al-Yateem 

Academic Editor

PLOS ONE